# Exploring the relationship between local food environments and obesity in UK, Ireland, Australia and New Zealand: a systematic review protocol

Andrea Fuentes Pacheco,[1] Gabriela Carrillo Balam,[1] Daryll Archibald,[2,3] Elizabeth Grant,[4] Valeria Skafida[5]

For numbered affiliations see end of article.

**Correspondence to**
Miss Andrea Fuentes Pacheco;
Andrea.Fuentes@ed.ac.uk

## ABSTRACT

**Introduction** Obesity is a global pandemic that affects all socioeconomic strata, however, the highest figures have been observed in the most disadvantaged social groups. Evidence from the USA and Canada showed that specific urban settings encourage obesogenic behaviour in the population living and/or working there. We aim to examine the evidence on the association between local food environments and obesity in the UK, Ireland, Australia and New Zealand.

**Methods** Six databases from 1990 to 2017 will be searched: MEDLINE (Ovid), Embase (Ovid), Scopus, The Cumulative Index to Nursing and Allied Health Literature (CINAHL), Applied Social Sciences Index and Abstracts (ASSIA) and Web of Science. Grey literature will also be sought by searching Opengrey Europe, The Grey Literature Report and relevant government websites. Additional studies will be retrieved from the reference lists of the selected articles. It will include cohort, longitudinal, case study and cross-sectional studies that have assessed the relationship between local food environments and obesity in the UK, Ireland, Australia and New Zealand regardless of sex, age and ethnicity of the population. Two researchers will independently select the studies and extract the data. Data items will incorporate: author names, title, study design, year of study, year exposure data collected, country, city, urban/rural, age range, study exclusions, special characteristics of study populations, aims, working definitions of food environments and food outlets, exposure and methods of data collection, outcomes and key findings. A narrative synthesis and a summary of the results will be produced separately for children and adults, according to the type of food exposure–outcome. All the selected studies will be assessed using The Quality Assessment Tool for Observational Cohort and Cross-Sectional Studies.

**Ethics and dissemination** This study will be based on published literature, and therefore ethical approval has not been sought. Our findings will be presented at relevant national and international scientific conferences and published in a peer-reviewed journal.

### Strengths and limitations of this study

► This is the first systematic review to analyse the evidence on the relationship between local food environments and obesity in the UK, Ireland, Australia and New Zealand and to compare the findings with the North American results of a previously conducted review.
► It will incorporate price as a variable of the food environment. An additional analysis of the available foodstuff quality (healthy and unhealthy) inside food outlets will be completed.
► The review will be conducted in four countries with high obesity figures.
► As we cannot cover the whole European and Oceanic regions, we selected four countries to represent both continents.

According to WHO, over 600 million people worldwide were obese in 2014.[2] The increased trend has particularly affected high and upper-middle-income countries that have reached a high industrialisation and urbanisation level. However, a growing number of low-income and middle-income countries are also dealing with this pandemic alongside the additional burden of malnutrition.[3–6] While the prevalence is rising across all the population segments, the highest figures have been observed in the most disadvantaged social groups.[5 6] Obesity is now a major global health challenge especially among those who suffer socioeconomic disparities across the globe.[5 6]

This condition, regarded in some literature as a disease,[7] has been described as one of the most complex health problems of modernity due to its multicausal aetiology.[7 8] Beyond individual determinants, the complex social and physical contexts in which individual behavioural decisions are made appear to strongly influence the outcome.[9 10] Based on this approach, Swinburn *et al* defined the

## INTRODUCTION

The obesity pandemic has been increasing dramatically on a global scale since 1980.[1 2]

**BMJ**

concept of the obesogenic environment as the 'sum of the influences that the surroundings, opportunities or conditions of life have on promoting obesity in individuals and populations'.[11] Understanding the environmental influences over people's eating behaviours is a major challenge for researchers because people live and function in multiple urban areas where there are many opportunities to shop and eat food throughout the day.[12] Though some studies have found a negative or null association between food environments and obesity,[13 14] other studies do point to evidence that deprived food environments encourage an obesogenic food behaviour in the population living and/or working there.[8 10 13 15 16] Studies from a residential perspective have described how these environments appear to encourage an excessive energy intake and weight gain in the medium and long term, especially in those residents whose mobility is limited because of health or transport access such as the elderly and those on low incomes, but more work needs to be done to better understand these associations and in particular if there are types of food, for example, high-energy-dense foods that are featuring excessively in diets.[8 10 13–16]

Local food environments include the social, macrolevel and physical aspects that influence people's food choices.[17] One factor related to the macrolevel dimension and two related to the physical aspects have been highlighted as key determinants for those who purchase food within deprived neighbourhoods: price, physical access and availability to (and quality of) foodstuffs inside food sources.[17 18] Food prices, which are mainly regulated by the governments and the global market, can become major barriers for low-income populations.[17 18] Drewnowski and other authors have identified that individuals under economic constraints, frequently shop and consume high-energy-dense foods which are generally cheaper than healthy products.[17 18] Physical access, which refers to the distance from households to food sources, has also been identified as a barrier for economically and physically disadvantaged people. Studies have described how such people often rely on the purchase of food in nearby and walkable areas rather than spending budgets on public or private transportation to purchase food further away.[13 17] Cummins, Gibson and Burgoine among others have shown that many deprived urban zones have a major density and exposure of less healthy food sources, increasing the access to high-energy-dense foods.[8 13 15 16 19–21] The in-store availability, depicted as the variety of food provision within food sources, is directly related to the quality of foodstuffs people can purchase. Provision of foods is different in affluent versus deprived areas, where in the latter the offer is frequently less healthy and less varied than in wealthier areas.[8 19–21] The intersection of these three determinants could facilitate obesogenic food purchases and food intake, leading to a steady increase in body fat over time.[8 17–21]

A large volume of evidence has been generated in North America on this topic during the last decade. According to Cobb *et al*, in spite of many studies having found a null or negative relationship, a substantial number of other studies have shown a positive association between the aforementioned food environment variables and the prevalence of obesity among children and adults.[13 17] In the case of Europe and Oceania, there is a lack of recent analysis of the emerging evidence despite the UK, Ireland, Australia and New Zealand being among the nations with the highest figures and the worst projections of obesity in Western Europe and Oceania.[4] Two important similarities between these countries and the North American scenario also lead us to believe that a discussion of the studies assessing this relationship is necessary. In all regions the obesity prevalence is concentrated in disadvantaged urban areas, and all of them are experiencing the same postnutritional transition with a strong influence of a globalised and industrialised food market.[6 18] Therefore this systematic review of the literature in the UK, Ireland, Australia and New Zealand is timely, alongside a comparison with the North American findings.

Eight systematic reviews and one scoping review, focusing mainly on the USA and Canada have examined the associations between environment and obesity.[10 13 14 22–26] Two analysed the whole built environment, including transportation and physical activity access.[14 25] Three explored the consumer retail food environments without a detailed analysis of the relationship with obesity.[10 23 24] Finally, one examined the association between the food environment and weight status and the other two the relationship with the obesity.[13 22 26] This will be the first systematic review to explore the relationship between local food environments and obesity in UK, Ireland, Australia and New Zealand, incorporating food price[11 18] as part of the food environment. An additional analysis of each variable will be carried out considering the quality of foodstuff (healthy and unhealthy) to which people are exposed in residential zones.

Cummins and Macintyre[9] recommend a regional analysis of the available research in other high-income countries in order to identify if the social, economic and geographical contexts are similar enough to attribute the same causes at a global scale.[9 10 26] Until now, the most consistent evidence for a 'contextual' effect of food environment is only available from North America.[9 13 14 26] Furthermore, following a variety of public health and private interventions created to promote healthy lifestyle within neighbourhoods over the last decade, such as the insertion of a higher number of supermarkets in deprived areas and social media campaigns about healthy eating,[27] it is important to discuss if and how this phenomenon is affecting the most vulnerable population living in these European and Oceanian countries mentioned above.

## Objectives

The primary objective is to examine the evidence on the association between local food environments and obesity in the UK, Ireland, Australia and New Zealand. The secondary objectives are to identify gaps in the evidence related to this particular association and analyse the

pertinence of this relationship considering the regional contexts.

## METHODS

We will carry out a systematic review of the literature. We will draw on the methodology developed by Cobb *et al* in 2015 who explored the relationship between local food environments and obesity in the USA and Canada.[13] The proposed review will extend the geographical scope of that work by focusing on studies conducted in the UK, Ireland, Australia and New Zealand. It will maintain the physical food access and availability dimensions included in the review by Cobb *et al* and will incorporate a third dimension: food price. Finally, it will be guided by The Preferred Reporting Items for Systematic Reviews and Meta-Analysis Protocols (PRISMA-P).[28]

### Eligibility criteria

#### Types of studies

All observational epidemiological studies that have assessed access and/or availability of food sources inside neighbourhoods (cohort, longitudinal, case study and cross-sectional) with group-level data and with individual-level data on more than 200 people will be included. It will exclude the studies with less than 200 people. This sample size threshold was used by Cobb *et al* who identified that studies with a smaller sample would be statistically underpowered for the detection of a significant association between the variables.[13] Our study will follow the same criterion as we wish to compare both reviews in a later discussion.

#### Participants

Eligible participants will include populations regardless of age, sex or ethnicity living in UK, Ireland, Australia and New Zealand. Studies with a separated analysis of adults and children will be included, and in the case where this information has not been provided, the authors will be contacted to request that specific data.

#### Years considered

The review will include studies from January 1990 to May 2017. The initial cut-point year was adopted by Cobb and other reviews, the rationale being that before the last decade of the 20th century very little data appeared in this field.[11 13 29]

#### Setting

The sample will include research articles only from UK, Ireland, Australia and New Zealand.

#### Language

Only articles written in English will be included as English is the predominant language of the selected countries.

#### Exclusion criteria

Literature exclusively looking at individuals with major pathologies, pregnant women, homeless populations, breastfeeding women and participants with physical limitations will be excluded. This is because these conditions independently affect nutritional status. Individuals with obesity grade 3 will also be excluded because this is the most severe stage of obesity and according to the evidence,[2 7] there are other physiological causes involved in that status (increases in morbid obesity).

### Search strategy

A preliminary scan in MEDLINE was carried out with the purpose of identifying and building a list of index and free terms (see online supplementary appendix 1). The final list of search terms was agreed through a consultative process with the review team, clinical and social science colleagues, and a senior librarian at The University of Edinburgh. Due to the iterative nature of the search process, additional search terms and sources may be incorporated into the search strategy. The following databases will be searched: MEDLINE (Ovid), Embase (Ovid), Scopus, The Cumulative Index to Nursing and Allied Health Literature (CINAHL), Applied Social Sciences Index and Abstracts (ASSIA) and Web of Science. Grey literature will also be sought by searching Opengrey Europe, The Grey Literature Report, relevant government websites related to the countries included in this review (UK Foresight Programme, Australian Institute of Health and Welfare, National library of Australia, PANDORA (Australian Government Web Archive) and Obesity Policy Coalition Australia) and other international organisations' websites related to this topic (World Obesity Federation, Spotlight Project and European Association of the Study of Obesity). Additionally, the reference lists of the selected articles will be checked for additional articles that can potentially be retrieved.

### Study records

#### Data management

Retrieved studies from databases, grey literature and hand-searching will be exported to Endnote Library. The programme will also be used for the screening and deduplication process.

#### Selection process

Two researchers (AFP and GCB) will independently undertake the selection of studies and data extraction. Discrepancies will be solved by consensus between the two authors. If required, a third party (EG) will make a judgement on the data entered and act as an arbitrator. Full text articles will be retrieved.

#### Data extraction

Data items were selected after a review of previous data collection strategies published in previous reviews[13 26] and through consideration of the variables this study seeks to explore through the evidence (see online supplementary appendix 2). Features to be extracted will include:
► authors' names
► article description

- ► design: title, study design, year of study, year exposure data collected, country, city, urban/rural, age range, study exclusions and special characteristics of study population
- ► aims and working definitions of food environments and food outlets
- ► exposure and methods of data collection
- ► outcomes reported: outcome definition, self-reported or measured
- ► statistical analysis
- ► key findings
- ► limitations.

Data from each study will be collated into the form, and a final database of all forms will be elaborated using a customised Excel sheet. The extraction form will be piloted before its full use in the review. During the pilot, the first 10 articles will be independently extracted and jointly reviewed by the two reviewers. The remaining articles will be extracted by one reviewer and reviewed for accuracy by the other.

### Outcomes and prioritisation

The primary outcome is obesity. The diagnosis of this pathology follows the criterion established by WHO to classify the obesity with a body mass index (BMI) over $30 \, \text{kg/m}^2$.[2] Change in BMI, as well as measurements of weight and height will be used to calculate and classify the nutritional status.

The secondary outcome is central obesity, represented by waist circumference and waist-to-hip ratio (only if the primary outcome is not included).

### Risk of bias in individual studies

All the selected studies will be assessed using The Quality Assessment Tool for Observational Cohort and Cross-Sectional Studies.[30] This checklist is a useful tool to develop a critical appraisal of these type of studies. The items contained are:
- ► research question
- ► study population
- ► sample size
- ► confounders
- ► selection bias
- ► data collection methods
- ► exposure and outcomes measures
- ► blinding of outcome assessors
- ► statistical analyses.

### Data synthesis

A separate narrative synthesis and a summary of the results for children and adults will be performed, according to the type of food exposure–outcome. The measurement techniques chosen in every study will also be analysed. Finally, this will be compared with the main findings generated in the original North American systematic review.

Subgroups analysis: men, women, adults, children, low socioeconomic status (SES), high SES, urban, rural and by country.

### Protocol registration

A detailed protocol was registered with the International Prospective Register of Systematic Reviews: 2017: CRD42017068193 (http://www.crd.york.ac.uk/PROSPERO/display_record.asp?ID=CRD42017068193). The study will be guided by the PRISMA-P 2015 statement.[28]

## CONCLUSIONS

Previous systematic reviews have assessed the evidence related to the association between local food environments and obesity in the USA and Canada.[13 26] This systematic review is the first study that will geographically extend the work undertaken by Cobb *et al*, analysing the relationship on evidence from the UK, Ireland, Australia and New Zealand. Furthermore, it will incorporate food price as part of the food environment and develop an additional analysis of the quality of foodstuff (healthy and unhealthy) available inside the residential areas measured. Finally, it will undertake a comparative analysis by country and between the regional results and the North American findings. This is highly relevant in order to gain a better understanding of whether the phenomenon is subject to regional variations or if it is occurring on a global scale.

## ETHICS AND DISSEMINATION

The review findings will be presented in relevant national and international scientific conferences and be published in a peer-reviewed journal.

**Author affiliations**
[1]Centre for Population Health Sciences, Usher Institute of Population Health Sciences and Informatics, The University of Edinburgh, Edinburgh, UK
[2]Scottish Collaboration for Public Health Research and Policy, Usher Institute of Population Health Sciences and Informatics, The University of Edinburgh, Edinburgh, UK
[3]School of Psychology and Public Health, College of Science, Health and Engineering, La Trobe University, Melbourne, Australia
[4]Global Health Academy, Usher Institute of Population Health Sciences and Informatics, The University of Edinburgh, Edinburgh, UK
[5]Social Policy, School of Social and Political Science, The University of Edinburgh, Edinburgh, UK

**Contributors** AFP conceived the idea for this work and drafted the protocol. The draft was critically revised according to several rounds of critical comments by DA, EG, VS and GCB. All the authors will be involved in the systematic review process.

**Funding** DA is supported jointly by the Medical Research Council (MRC; MR/K023209/1) and the Chief Scientist Office (CSO).

**Competing interests** None declared.

**Patient consent** Not required.

**Ethics approval** As this is a review of published literature ethics approval has not been sought. However, this work is subject to Institutional Review Board oversight by The University of Edinburgh's Centre for Population Health Sciences.

**Provenance and peer review** Not commissioned; externally peer reviewed.

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
