## [Reviewer comments · BMJ Open]

ARTICLE DETAILS

TITLE (PROVISIONAL)	Exploring the relationship between local food environments and obesity in UK, Ireland, Australia and New Zealand: A systematic review protocol
AUTHORS	Fuentes Pacheco, Andrea; Carrillo Balam, Gabriela; Archibald, Daryll; Grant, Elizabeth; Skafida, Valeria

VERSION 1 – REVIEW

REVIEWER	Margaret Allman-Farinelli University of Sydney Australia
REVIEW RETURNED	28-Aug-2017

GENERAL COMMENTS	This study would appear to be repeating the methods used by Cobb in North America but applying it to populations in the UK, New Zealand and Australia. (It is noted Ireland is to be included on page 8.) Given the high rates of obesity in these countries this appears justified. The authors suggest that sub-analyses will be conducted for gender, SES and more. These are very important but the most important differences may be by country because of the diversity of ethnicity in the populations eg it is well known in New Zealand that Maori and Pacific Island populations are much more likely to become obese and in Australia we know that children from Middle-Eastern backgrounds are more likely to become obese. In parts the English expressions used in the manuscript are not perfect and I strongly suggest that all the authors take another read of the paper to correct this. eg line 34 in the abstract "It will be included cohort, "eg line 17 of page 7 . The review indicates it will consider not only food availability and accessibility but also cost of food. The variables that can be collected will only be as good as the studies that have been published but it will be interesting to see how many include this variable. There is also discussion of including data on healthy versus unhealthy food. I am assuming the authors might be expecting studies that report this variable. Traditionally fast food stores and quick service restaurants have been viewed as unhealthy and supermarkets more healthy as proxies for actually assessing individual foods on sale. One of the difficulties of these types of studies is that people often travel far from their local food environment and may even purchase all their meals and snacks in locations close to where they work rather than the local food environment.
--

	This is a basic flaw in most studies. Perhaps this could be mentioned in the conclusion here. It is stated that observational studies will be the major study design - yet the tool to assess the quality of the studies appears to be one designed for RCTs. It is suggested the authors find an alternative tool. Cochrane is currently developing a tool for observational studies. I am somewhat uncertain as to whether an additional protocol to that published in PROSPERO is needed.
--	---

REVIEWER	Julia Diez Social and Cardiovascular Epidemiology Research Group, School of Medicine, University of Alcala, Spain.
REVIEW RETURNED	29-Aug-2017

GENERAL COMMENTS	The authors have addressed an interesting topic within the food environment research: the need of studying the link between food environments and obesity outside the US and Canada. However, before considering it for publication, and prior to the systematic review being conducted, some major changes should be made. Specific comments Introduction Overall, the introduction is well-written, but needs to be more thorough in its examination of the current literature. Some minor improvements could be made in: Lines 51-54: it would be helpful if the authors could rewrite these lines explaining the dimensions of the local food environment. Consider using a conceptual framework to better structure this paragraph. A useful article would be Story M., Kaphingst K., Robinson-O'Brien R., Glanz K. 2008. Creating Healthy Food and Eating Environments: Policy and Environmental Approaches. Annual Review of Public Health. While the authors explain the results of previous reviews, it would be helpful to include other seminal articles in the introduction (e.g. Feng J, Glass TA, Curriero FC, Stewart WF, Schwartz BS. The built environment and obesity: a systematic review of the epidemiologic evidence. Health Place 2010; Giskes K, van Lenthe F, Avendano-Pabon M, Brug J. A systematic review of environmental factors and obesogenic dietary intakes among adults: are we getting closer to understanding obesogenic environments? Obes Rev 2010; or Holsten JE. Obesity and the community food environment: a systematic review. Public Health Nutr 2009. Methods Participants: It would be helpful if authors could briefly explain in the manuscript what they would do if the studies addressing both adult and children do not provide the data separately. Will these studies be included? Will the authors be contacted for the specific data? Data extraction: The data extraction form should be included in the review protocol as an appendix or as online supplementary material. Moreover, the authors could briefly describe the training and the piloting of the extraction form.
---

	Data items: It would be helpful if authors could briefly explain in the manuscript what were the data they extracted on the exposure (local food environment), and how were these chosen. Outcomes: The authors may want to add a line in the text explaining why they do not include overweight as a secondary outcome. Conclusions This section needs to build more on what this review can add, that hasn't already been added by prior reviews in this field It would be helpful if the authors could briefly explain the potential limitations of their review; and how will they disseminate the results of their study.
--	--

REVIEWER	Fidelia Dake, Ph.D. Regional Institute for Population Studies University of Ghana Ghana
REVIEW RETURNED	11-Sep-2017

GENERAL COMMENTS	The authors of this manuscript propose to conduct a systematic review on the relationship between the local food environment and obesity in the UK, Ireland and New Zealand. This is a relevant topic that will make useful contributions to the literature on the subject matter. The authors however fail to present convincing arguments in the manuscript. The manuscript is poorly written and there a lot of grammatical errors. The overall structure, and language of the manuscript need to be improved before it can be considered for publication. For example the second and third sentence of the ethics and dissemination statement is a repetition. The authors should cross-check this. The references provided are generally ok but it can still be substantially improved. some references example no. 11 Jensen MD et al. needs to be written in full with the names of the authors. Also some key literature on the subject of local food environments and obesity e.g. A book title Geographies of Obesity is not included in manuscript. Another paper by Papas et al. (2007) on the built environment and obesity is also not mentioned in this manuscript. Furthermore the end-text reference list is not accurately written.
---

VERSION 1 – AUTHOR RESPONSE

REVIEWER #1

Reviewer Name: Margaret Allman-Farinelli

Institution and Country: University of Sydney, Australia

Competing Interests: Attached

Comment 1: The authors suggest that sub-analyses will be conducted for gender, SES and more. These are very important but the most important differences may be by country because of the diversity of ethnicity in the populations

Response: We have included a sub-analysis by country (page 14, line 321-322).

Comment 2: In parts the English expressions used in the manuscript are not perfect and I strongly suggest that all the authors take another read of the paper to correct this.

Response: We read the manuscript carefully and have corrected all the grammatical and typographical mistakes. We also improved the English expressions, including the given example in line 74 (abstract): "It will be included cohort..." we rewrite "It will include cohort..."

Comment 3: One of the difficulties of these types of studies is that people often travel far from their local food environment and may even purchase all their meals and snacks in locations close to where they work rather than the local food environment. This is a basic flaw in most studies. Perhaps this could be mentioned in the conclusion here.

Response: We thank the reviewer for highlighting this point. We firmly agree with the comment, and this relevant issue will be considered in the review discussion. Given that the conclusions section is brief and we have a limited space to summarize the study relevance and justification, we have decided to introduce some lines related to people 'mobility and the challenge that this means in the background section (page 5, lines 125-128). The place-based studies limitations will be intensely discussed in the review manuscript.

Comment 4: It is stated that observational studies will be the major study design - yet the tool to assess the quality of the studies appears to be one designed for RCTs. It is suggested the authors find an alternative tool. Cochrane is currently developing a tool for observational studies.

Response: We thank the reviewer for raising this important issue. We reconsidered this point and agreed to change the instrument. Considering that the sample of previous reviews was mainly composed of observational studies, it will apply now The Quality Assessment Tool for Observational Cohort and Cross-Sectional Studies. We believe this tool is more appropriate to analyse our sample, particularly as it covers essential items of the critical appraisal.

Comment 5: I am somewhat uncertain as to whether an additional protocol to that published in PROSPERO is needed.

Response: We would like to publish this protocol to keep the transparency of the review design, as well as to provide detail on the methodology of the study. PROSPERO registration contains just a summary of the protocol and informs partially the aspects mentioned above.

REVIEWER #2

Reviewer Name: Julia Diez

Institution and Country: Social and Cardiovascular Epidemiology Research Group, School of Medicine, University of Alcala, Spain

Competing Interests: None declared

Introduction

Comment 1: Lines 51-54: it would be helpful if the authors could rewrite these lines explaining the dimensions of the local food environment. Consider using a conceptual framework to better structure this paragraph. A useful article would be Story M., Kaphingst K., Robinson-O'Brien R., Glanz K. 2008. Creating Healthy Food and Eating Environments: Policy and Environmental Approaches. Annual Review of Public Health.

Response: We thank the reviewer for pointing out this important aspect and for the bibliography suggested. The article provided a useful framework that led us to enrich the food environment description (page 6, lines 137-149 and lines 155-157). We classified the food environment dimensions according to the author's proposal and have better explained each one considering the cited evidence. (Annu Rev Public Health. 2008;29:253-72).

Comment 2: While the authors explain the results of previous reviews, it would be helpful to include other seminal articles in the introduction (e.g. Feng J, Glass TA, Curriero FC, Stewart WF, Schwartz BS. The built environment and obesity: a systematic review of the epidemiologic evidence. Health Place 2010; Giskes K, van Lenthe F, Avendano-Pabon M, Brug J. A systematic review of environmental factors and obesogenic dietary intakes among adults: are we getting closer to understanding obesogenic environments? Obes Rev 2010; or Holsten JE. Obesity and the community food environment: a systematic review. Public Health Nutr 2009.

Response: We thank the reviewer for suggesting these additional resources. We reviewed the resources and incorporated some of them into the background (page 5, lines 120-122 and 128-131; page 6, lines 131-136) and in the summary of the reviews that have been conducted on this topic (page 7, lines 172-173 and lines 176-177)

Methods

Comment 3: It would be helpful if authors could briefly explain in the manuscript what they would do if the studies addressing both adult and children do not provide the data separately. Will these studies be included? Will the authors be contacted for the specific data?

Response: We have now included the suggestion in our inclusion criteria section (page 9, lines 222-224).

Comment 4: The data extraction form should be included in the review protocol as an appendix or as online supplementary material. Moreover, the authors could briefly describe the training and the piloting of the extraction form.

Response: We have now included a preliminary extraction form as a Supplementary Appendix. We also briefly described the piloting process of the Word extraction form (page 12, lines 288-291).

Comment 5: It would be helpful if authors could briefly explain in the manuscript what were the data they extracted on the exposure (local food environment), and how were these chosen.

Response: We have now included an explanation of the selection process of the food environment data that will be extracted in the review (page 12, lines 272-274).

Comment 6: The authors may want to add a line in the text explaining why they do not include overweight as a secondary outcome.

Response: We thank the reviewer for this suggestion, and we agree to consider overweight as a secondary outcome as a pre-stage of the obesity. However, as we are aiming to establish a

continuum in the literature we have followed the same and the main and secondary outcomes as other major reviews.

Conclusion

Comment 7: This section needs to build more on what this review can add, that hasn't already been added by prior reviews in this field

Response: We have built up this section, focusing in the review on the strengths and highlighting its future contributions (page 14, lines 333-341)

Comment 8: It would be helpful if the authors could briefly explain the potential limitations of their review; and how will they disseminate the results of their study.

Response: We are aware this review has limitations which will be carefully described in the review manuscript. We included the dissemination plan in the abstract (page 4, lines 88, 89) and at the end of the text (page 15, lines 346,347)

REVIEWER #3

Reviewer Name: Fidelia Dake, Ph.D.

Institution and Country: Regional Institute for Population Studies, University of Ghana, Ghana

Competing Interests: None declared

Comment 1: The authors however fail to present convincing arguments in the manuscript.

Response: We appreciate this perspective but do believe that the arguments within the paper do present a valuable contribution to the discussion. We have however strengthened the justification on why this review should be conducted (page 7, lines 165-171; page 7, lines 178-182) and improved the background, in order to facilitate the reader, the understanding of this topic (page 5, lines 117-118 and lines 125-132; page 6, lines 137-151).

Comment 2: The manuscript is poorly written and there a lot of grammatical errors. The overall structure, and language of the manuscript need to be improved before it can be considered for publication. For example the second and third sentence of the ethics and dissemination statement is a repetition. The authors should cross-check this.

Response: We read the manuscript carefully and corrected all the grammatical and typographical mistakes, including the paragraph in the ethics and dissemination section (page 15, lines 346,347).

Comment 3: The references provided are generally ok but it can still be substantially improved. some references example no. 11 Jensen MD et al. needs to be written in full with the names of the authors. Furthermore, the end-text reference list is not accurately written.

Response: We have carefully cross-checked and properly completed all the references (pages 16-19).

Comment 4: Also some key literature on the subject of local food environments and obesity e.g. A book title Geographies of Obesity is not included in manuscript. Another paper by Papas et al. (2007) on the built environment and obesity is also not mentioned in this manuscript.

Response: We thank the reviewer for suggesting these resources. We have incorporated the article "The built environment and obesity" in the summary of the reviews conducted on this topic. Definitely

the book entitled "Geographies of Obesity: Environmental understandings of the obesity epidemic" and will be incorporating in the review manuscript.

VERSION 2 – REVIEW

REVIEWER	Margaret Allman-Farinelli University of Sydney Australia
REVIEW RETURNED	03-Dec-2017

GENERAL COMMENTS	The authors have addressed most of the concerns. Some minor points for consideration are included. Abstract I do not understand what you are doing with the methods section. The track changes section has removed the first two sentences but the clean PDF is fine. Objectives page 8. I do not believe you have satisfactorily addressed reviewer two's suggestion about including overweight as a secondary outcome. You say you have made it an outcome but this is not at all apparent in the objectives. Overweight in children is most certainly indicative of the development of obesity as an adult.
---

VERSION 2 – AUTHOR RESPONSE

REVIEWER COMMENTS:

Reviewer Name: Margaret Allman-Farinelli
Institution and Country: University of Sydney, Australia
Competing Interests: Attached

Comment 1: I do not understand what you are doing with the methods section. The track changes section has removed the first two sentences but the clean PDF is fine.

Response: We have already corrected that mistake in the marked copy (page 3)

Comment 2: I do not believe you have satisfactorily addressed reviewer two's suggestion about including overweight as a secondary outcome. You say you have made it an outcome but this is not at all apparent in the objectives. Overweight in children is most certainly indicative of the development of obesity as an adult.

Response: We thank the reviewer for pointing out this important issue and apologise for any ambiguity in the original reply. We agree overweight is a pre-stage to obesity, however, after consideration, we decided not to include this in the analysis as we are aiming to establish a continuum in the literature using the same outcomes used by Cobb et al. (Obesity (Silver Spring). 2015;23(7):1331–44) and other reviews. Furthermore, we consider that these outcomes (obesity and central obesity) are enough to achieve the review objectives.